# Health professionals' competence for the provision of quality primary health care in Amhara region, Ethiopia

**Gebeyehu Tsega**[1,2]*, **Mirkuzie Woldie**[1,2,3], **Gizachew Yismaw**[4], **Getu Degu**[5]

**1** Department of Health Systems Management and Health Economics, School of Public Health, College of Medicine and Health Sciences, Bahir Dar University, Bahir Dar, Ethiopia, **2** Fenot Project, Department of Global Health and Population, Harvard T.H. Chan School of Public Health, Addis Ababa, Ethiopia, **3** Department of Health Policy and Management, Faculty of Public Health, Jimma University, Jimma, Ethiopia, **4** Health Research Development Directorate, Amhara Public Health Institute, Bahir Dar, Ethiopia, **5** Department of Epidemiology, School of Public Health, College of Medicine and Health Sciences, Bahir Dar University, Bahir Dar, Ethiopia

* gebishts@gmail.com

## Abstract

### Background

Though competent health professionals are essential for building strong and resilient health systems; there is a dearth of evidence on whether health professionals possess core competencies for providing quality primary health care in Ethiopia. Therefore, the aim of this study was to examine health professionals' competence in the provision of quality primary health care in Amhara region, Ethiopia.

### Methods

A mixed methods study design with pragmatic philosophical paradigm was conducted on, 846 (for quantitative) and 12 (for qualitative) selected, health professionals from June 1–July 30/2023. Health professionals' competence was measured through six domains, adapted from the World Health Organization (WHO) global competency framework for universal health coverage. Quantitative and qualitative data were collected. Logistic regression modeling and thematic analysis were carried out.

### Results

The response rate was 98%. As rated by themselves, only 116 (14%) health professionals were competent for all competencies. The rating for specific competency domains was slightly higher with a range of 21.7% (180) to 30.7% (255) of the professionals were competent in personal conduct and evidence informed practice domains, respectively. The qualitative findings support the competence gaps identified in health professionals' survey. Educational status, training, taking licensure/ certificate of competence (COC) exam, training in public universities/colleges, high cumulative GPA and monthly salary above 10,000 ETB (177.84$) positively affected the rating of the competence.

**Data availability statement:** All relevant data are within the manuscript.

**Funding:** The author(s) received no specific funding for this work.

**Competing interests:** The authors have declared that no competing interests exist.

## Conclusions

The rate of health professional competence as judged by the health professionals themselves in the study area was very low. The qualitative findings also identified several competence problems. Progressive health professional development in the form of upward and in-service training, provision of licensure/COC exam, and learning in public universities/colleges positively impact professional competence. Therefore, the health and education systems together should strengthening upgrading and in-service training including CPD; licensure/COC exam; optimize the salary and strong regulation of private colleges.

## Introduction

Health professionals' competence refers to their ability to provide quality healthcare that is effective, safe, people-centered, timely, equitable, integrated, and efficient. Health professionals, equipped with requisite competencies, are essential for building strong and resilient health systems to achieve global and national health goals and targets, including universal health coverage (UHC) and health security through primary health care (PHC) lens [1–3]. As reflected in the two United Nation (UN) high level meetings on UHC (2019 and 2023) [4,5] and in the two international declarations on PHC at Alma-Ata(1978) and Astana (2018) [3,6], PHC with competent health professionals, among others, is recognized as the most effective, efficient, and equitable approach to ensure that no one is left behind [3]. With this in mind, recently, the WHO has developed the PHC measurement framework in 2021 [3] and the respective global competency framework for UHC in 2022 [1]. This competency framework defines the core competencies that all health professionals need to provide quality PHC. It has six domains: (1) people-centeredness, (2) decision-making, (3) communication, (4) collaboration, (5) evidence-informed practice and (6) personal conduct [1,7].

As such, health professionals' competence is a global, national and subnational policy priority to ensure that citizens have access to quality healthcare [8–17]. For example, globally, WHO acknowledged that no health without competent health professionals [12] and has committed to addressing health professionals' competence issues by endorsing the "Global Strategy on Human Resources for Health 2030". This global strategy aims to optimize the competence of health professionals, among others, for the provision of quality health care [10]. Similarly, in Ethiopia, health professionals' competence is a policy priority as reflected in the current policies and strategic plans at national and subnational levels with special focus on PHC that typically serves as the gatekeeper role in the health system [14–19]. For example, the Ethiopian Ministry of Health (EMOH) developed different strategic documents such as strategic plan for health professionals' competency assessment and licensing, a roadmap for continuing professional development (CPD) for health professionals and implementation strategy for motivated, competent and compassionate (MCC) health professionals to improve the competence of health professionals at national and subnational levels [20–22].

To implement these strategic plans and policies, Ethiopia has introduced various initiatives to enhance the competence of health professionals in a sustainable manner. These include adopting global initiatives that promote competency-based, information technology-facilitated, and inter-professional education for health professionals and optimizing the quality of existing health professionals through needs-based continuous professional development (CPD) and lifelong learning [7,23,24].

In response to the development of new educational technologies and the unprecedented amount of change in health-care systems, Ethiopia aspires to transform health professionals'

education with the aim of addressing health professionals' competence gap through strong collaboration of education and health systems [15,22,25]. For example, in Ethiopia and elsewhere, while students are in educational institutions, existing jobs are being replaced by new ones, making it impossible for new graduates to learn all the competencies required in the workplace. This problem can be addressed through using the opportunity of new educational technologies that makes health professionals develop competencies at anytime and anywhere through adapting the education-for-life model in higher education and health care setting [1,7].

Despite the efforts made, there is a growing concern that the current education system in Ethiopia does not adequately prepare health professionals to solve the complex health challenges of the 21st century. This is due to various reasons, including the flooding education strategy of Ethiopia [26–28]. Moreover, one of the main concerns in primary health care settings in Ethiopia and Amhara region is a poor quality of health care, both now and in the future [29,30]. The low quality of health care at PHC facilities may be caused by the low competence of health professionals working there.

Few prior studies have applied particular competency frameworks to examine the competence of either specific segments or certain groups of health professionals, mainly doctors and nurses [31–34]. However, there is a dearth of evidence on whether health professionals possess core competencies for providing quality primary health care in Ethiopia. Evidence on such health professionals' competence and its associated factors is critical for policy/decision makers and implementers to design and implement strategies to address competence problems. Therefore, the aim of this study was to examine health professionals' competence (through WHO global competency framework for UHC) in the provision of quality primary health care in Amhara region, Ethiopia.

## Methods

### Study design and setting

A multi-centered cross-sectional study design, employing concurrent mixed methods approach with pragmatic philosophical paradigm, was conducted from June 1-July 30/2023 in Amhara region. Amhara region is one of the most populous regional states in Ethiopia; it has 14 zones, 8 city administrations and 180 districts (139 rural and 41 urban). According to the Ethiopian Central Statistics Agency, the region has a projected population of 30.9 million. About 80 percent of them are rural dwellers. The region has 100 hospitals, 917 health centers, and 3,725 health posts. There are 32,589 health professionals (40% of whom are females) in Amhara region. From which, 36.8%, 13.4%, 10.4%, 4% are nurses, midwives, health officers and doctors, respectively [35,36].

### Study participants

Randomly selected health professionals who are employed in primary hospitals and health centers in Amhara region were study participants of the quantitative data, whereas purposively selected experienced health professionals were the study participants of the qualitative data.

### Variables.

**Dependent variable.** Health professionals' competence

**Independent variables.**

**Health professionals' related factors:** educational background, age, sex, experience, taking in-service training (IST), field of study, salary, marital status, time management, having a vision, curiosity, having mentor, having license, take licensure/COC exam.

**Training institution related factors:** being public/private higher education institutions; experience in team training program; seniority of higher education institutions; curriculum being implemented (conventional/competency based); perceived availability of standard skill lab; perceived availability of standard class rooms; perceived instructor s' competence during their training, perceived availability ICT infrastructure and library with latest and relevant books/ soft and or hard copy; and perceived supportive academic leaders.

**Health facilities related factors:** merit based promotion/recognition; seniority of health facility(old/new), type of health facility (health center versus primary hospital); availability of library with latest and relevant books; merit based evaluation, availability of experienced professional; supportive health workforce leadership, competency based recruitment; availability of ICT infrastructure, organizational culture.

**Other factors:** COVID-19 (graduated after outbreak of COVID-19) and conflict (graduated during the conflict).

Moreover, perceived health professionals' competence was the variable for qualitative data.

## Sample size determination

The sample size is calculated through single population formula with assumptions: proportion of 50%, 95% CI, and margin of error of 5% [37].

$$n = \frac{Z(\alpha/2)^2 * P(1-p)}{d^2}$$

Where, P = 50% (proportion of competent health professionals)
  d = 0.05 (margin of error) and Z α/2 at 95% confidence level = 1.96
  By taking the above values, the sample size is

$$n = \frac{(1.96)^2 * (0.5)\,(1-0.5)}{(0.05)^2} \approx 384$$

Adding the 10% of non-response rate [39] with design effect of 2, the sample size was 846 health professionals. Sample sizes for factors were estimated and they were less than 846. Hence the sample size for the quantitative data was 846. Twelve experienced health professionals were participated in the qualitative data based on information saturation.

## Sampling methods

A multi-stage sampling method was used. First 3 zones (West Gojam, South Wollo and South Gondar) and 3 town health offices (Debre Tabor, Kobolecha and Dessie), were randomly selected. From the selected zones and towns, 5 primary hospitals and 40 health centers, a total of 45 health facilities, were selected randomly and 846 health professionals were selected proportionally from the selected health facilities by using simple random sampling method. The list of health professionals were obtained from human resources registry in each health facilities. Twelve experienced health professionals who have rich information about health professionals' competence were selected purposively for the in-depth interview.

**Data collection instruments and measurements.** The data collection instrument builds on previous literature and adapted from the WHO competency framework for universal health coverage and Dreyfus model [1,38,39]. The WHO competency framework for universal health coverage (UHC) builds on previous frameworks such as the CanMEDS and integrates

new emerging competencies for health professionals, such as critical thinking, systems thinking, lifelong learning, people centredness, and creativity. The aim of this framework is to prepare health professionals to provide quality PHC and achieve UHC that can be used across all health professionals and countries. The data collection instrument for quantitative data included sociodemographic variables, training institutions and health facilities related variables, and health professionals' competence variables. The in-depth interview guide was used for the qualitative data.

The outcome variable (health professionals' competence) was measured through six competency domains (people centeredness, decision making, communication, collaboration, evidence informed practice, and personal conduct) with 30 competencies adapted from WHO competency framework for universal health coverage and Dreyfus model. Each competency statement was rated by health professionals through three stages (1-not capable, 2-competent, 3-proficient) [38, 39]. If a health professional rated 2 and above (competent or proficient) in all items/competencies, he/she is competent (coded 1), otherwise incompetent (coded 0). The stages and the competency domains were defined as follow:

- **Not capable:** The health professional is unable to perform the minimum required tasks according to own judgment.

- **Competent:** The health professional can perform fully at least the minimum required tasks according to own judgment.

- **Proficient:** According to own judgment, the health professional is able to take full responsibility for own work and that of others where applicable and/or creating own interpretations.

**People-centredness:** includes 4 competencies related to the provision of health services that incorporate perspectives of individuals, caregivers, families and communities as participants in and beneficiaries of health systems. If a health professional rated 2 and above in the 4 competencies, he/she is competent in people centeredness.

**Decision-making:** includes 5 competencies related to the approach to decision-making. If a health professional rated 2 and above in the 5 competencies, he/she is competent in decision making.

**Communication:** includes 5 competencies related to effective communication. If a health professional rated 2 and above in the 5 competencies, he/she is competent in communication.

**Collaboration:** includes 4 competencies related to the practice philosophy of teamwork. If a health professional rated 2 and above in the 4 competencies, he/she is competent in collaboration.

**Evidence-informed practice:** includes 3 competencies related to the generation of evidence and information and their integration into practice. If a health professional rated 2 and above in the 3 competencies, he/she is competent in evidence informed practice.

**Personal conduct:** includes 9 competencies related to self-governed behaviors. If a health professional rated 2 and above in the 9 competencies, he/she is competent in personal conduct.

**Health professional:** implies front-line health professionals providing services targeted to patients and/or populations such as doctors, nurses, midwives and public health officers, laboratory, pharmacy [40,41].

## Data collection procedures

Nine trained data collectors with BSc degrees and three supervisors with master's degrees were recruited to collect the data. Before starting the data collection, clear explanations about the purpose of the study and the way how they filled the questionnaire were given for each study participant. Then, the data were collected through self-administered,

structured-questionnaire. Health professionals who were not present at their health facilities at the time of data collection were not eligible for the study. Health facilities related data were collected through direct observation and interviewing the managers of the health facilities for the quantitative data. In-depth interviews, with an average duration of 50 minutes, were conducted using a semi-structured questionnaire among experienced health professionals. These interviews were recorded and noted by data collectors and the principal investigator, all of whom had experience in qualitative data collection. The purpose was to triangulate the findings with the health professionals' self-reported competence.

## Data quality assurance

To ensure the quality of the data, care was taken throughout the data collection process, from prior to data collection to entry and analysis. The tool was validated using the Delphi method, and data collectors and supervisors received three days intensive training on the entire data collection process. Before the actual data collection, the data collectors carried out a role-play practice on data collection procedures during training. Questionnaire understandability, interviewing techniques, and all appropriate data collection procedures were tested on 5% of the sample size of health professionals. Based on the pretest findings, all necessary corrections were considered before fieldwork. To minimize social desirability bias, a self-administered questionnaire method was used for health professionals' surveys. Completeness of data and data cleaning (after data collection) were undertaken. Moreover, attention was also given during data coding, entry, and processing. Lastly, assumptions were checked to reduce statistical errors while fitting the statistical models.

Trustworthiness of the qualitative data was assured as per Guba and Lincoln's trustworthiness criteria (credibility, transferability, dependability, and confirmability) [42]. To ensure credibility of data, techniques such as prolonged engagement, triangulation, and member checking were used. Transferability was assured by thick description and heterogonous purposive sampling. Dependability was assured through code book, audit trail, peer debriefing and negative case analysis. Confirmability was assured by reflexivity, an audit trail, and peer debriefing. Audio taped data transcriptions and translations were made; the accuracy of the transcripts was continuously crosschecked against the audio recordings.

## Data management and analysis

Data completeness and missing data were checked and treated accordingly before data entry to Epi-data. Data entry, data codding, cleaning were carried out. Outliers and assumptions were checked to perform transformation before analysis. Both descriptive and analytical statistics with SPSS-25 were used. Frequency tables, mean, standard deviation and range were used to describe and summarize the data. Initially, multi-level logistic regression was planned, however, in the actual data; there is no variation at health facility level. The independence assumption was fulfilled; hence, binary logistic regression model was performed to identify the predictors of health professionals' competence. First, simple logistic regression was carried out to identify candidate variables for multiple logistic regressions. Variables with a significance level of p-value of <0.2 in simple logistic regression model were entered into the multiple logistic regression model to control for confounders. The overall adequacy of model was assessed using the Hosmer and Lemeshow test. All assumptions of logistic regression were checked and satisfied.

Regarding qualitative data analysis, the authors took several steps to ensure thorough and accurate interpretation. They began by familiarizing themselves with the data through careful and frequent listening to the audio recordings, followed by transcription and reading the

transcriptions multiple times. Then, translation was done. The authors generated codes and identified themes based on patterns and relationships within the data. To facilitate the analysis, they used ATLAS.ti.9, a qualitative data analysis software. Content analysis was conducted to identify the main themes and sub-themes. Additionally, they triangulated the qualitative data with the quantitative data to ensure comprehensive analysis and validation.

## Ethical approval and consent to participate

All methods used in this study were in accordance with the latest Ethiopian health research national ethics guideline (2022) and Declaration of Helsinki. The proposal was reviewed (by IRB) as per the standard procedure (SOP) of the guideline. Then, ethical clearance was obtained from Institutional Review Board (IRB) of Bihar Dar University with a protocol number of 705/2023 on March 06/2023. A formal letter, from the school was submitted to each concerned bodies to obtain their co-operation. Explanatory letter was added to each questionnaire to maintain participants' rights. All participants asked to participate in the study and received full explanations about the research purpose. Respect, anonymity and confidentiality were given and maintained by consent form for each participant. The liberty of participants to withdraw at any stage of the interview was maintained. Then, written informed consent was obtained from the participant as per the Institutional Review Board (IRB) approval.

## Results

### Sociodemographic characteristics of health professionals

Eight hundred thirty (830) health professionals participated in the study with a response rate of 98%. From which, 465 (56%), 715 (86.1%) and 405 (48.8%) were male, married and nurses, respectively. From the participants, 551 (66.4%) and 532 (64.1) were degree holders and graduated from public higher education institutions, respectively. Regarding to work experience, 590 (71.1%) of the health professionals had above five years of experience and 730(88%) of participants were working in health centers. The age of health professionals ranged from 19–50, with a mean of 31.62(standard deviation of 6.16), years. The monthly salary ranged from 3333 ETB (59.31$)-11305(201.16$), with a mean of 7079.51 (standard deviation of 1881.7), Ethiopian Birr. Most of the health professionals, 665(80.1%), were young, with below 35 years (Table 1).

### Health facility observation

During observation and interview of the health facility managers, we found that of the 40 health centers and 5 primary hospitals, none of them had functional library with relevant resources (books, internet, computers, tables, and chairs). Not a single study facility had a plan for and/or implemented continuous professional development (CPD) at the time of data collection. Disciplines of learning organizations (shared vision, systems thinking, mental models, team learning, and personal mastery), merit/performance-based evaluation and recognition, supportive health facility leaders, culture of communicating job descriptions for health professionals were absent or partially existed in the health facilities. Moreover, learning forum (e.g., morning session) does not exist in the health centers and was infrequently practiced in the primary hospitals.

**Health professionals' competence.** The health professionals' survey revealed that only 116 (14%) health professionals rated themselves as competent for all competency domains. The level of self-reported professional competence in the six domains ranged from 21.7% (180) to 30.7% (255) in personal conduct and evidence-informed practice domains, respectively. Fewer than one third of health professionals were found to be competent in each domain. A

**Table 1. Sociodemographic characteristics of health professionals, Amhara Region, Ethiopia, 2024 (n = 830).**

| Variables | Frequency | Percent |
|---|---|---|
| Sex | | |
| Male | 465 | 56 |
| Female | 365 | 44 |
| Age | | |
| <35 | 665 | 80.1 |
| 35–44 | 105 | 12.7 |
| >45 | 60 | 7.2 |
| Marital status | | |
| Married | 715 | 86.1 |
| Single | 115 | 13.9 |
| Educational status | | |
| Degree | 551 | 66.4 |
| Diploma | 279 | 33.6 |
| Learning institution | | |
| Public | 532 | 64.1 |
| Private | 298 | 35.9 |
| Profession | | |
| Nurse | 405 | 48.8 |
| Health officer | 235 | 28.3 |
| Midwife | 130 | 15.7 |
| Laboratory | 30 | 3.6 |
| Pharmacy | 20 | 2.4 |
| Doctor | 10 | 1.2 |
| Work experience | | |
| <2 years | 70 | 8.4 |
| 2–5 years | 170 | 20.5 |
| >5 years | 590 | 71.1 |
| Type of working institution | | |
| Health center | 730 | 88 |
| Primary hospital | 100 | 12 |
| Monthly salary | | |
| <5000(88.9$) | 130 | 15.7 |
| 5000–10,000(88.9$–177.84$) | 600 | 72.3 |
| >10,000(177.84$) | 100 | 12 |

higher proportion of health professionals, 255 (30.7%), were competent in evidence-informed practice, while a lower proportion, 180 (21.7%), were competent in personal conduct. The ratings for specific competency domains were slightly higher than the overall competence rating (Table 2).

Similarly, in the in-depth interviews, most interviewed health professionals reported that providing competent care in the context of primary health care (with low clients' health literacy, overwhelmed health providers and limited resources) does not exist in the real world. A 43-year health professional with a work experience of 19 years reported that "*You are currently unable to provide quality health care services to all patients in your health center due to heavy workload and limited capabilities and resources. As a result, you are forced to compromise some parameters of quality health care, including client-centeredness. You want to prioritize the*

Table 2. Health professionals' competence for the provision of quality PHC, Amhara region, Ethiopia, 2024.

| Domains | Level of competence | |
|---|---|---|
| | Competent-frequency (%) | Incompetent-frequency (%) |
| People-centredness | 210 (25.3%) | 620 (74.7%) |
| Decision-making | 195 (23.5%) | 635 (76.5%) |
| Communication | 215 (25.9%) | 615 (74.1%) |
| Collaboration | 200 (24.1%) | 630 (75.9%) |
| Evidence-informed practice | 255 (30.70%) | 575 (69.3%) |
| Personal conduct | 180 (21.7%) | 650 (78.3%) |
| All competencies | 116 (14%) | 714 (86%) |

*principle of 'first do no harm' during the provision of primary health care services, but you are overwhelmed with your heavy workload."*

Almost all health professionals expressed the need to improve their competence gaps through reflections, trainings, and lifelong learning. However, they identified the absence of various platforms, including digital ones, and a library with relevant resources as critical constraints for developing and updating their competence at their workplace. A 44-year health professional with 22 years of work experience reported that *"We [health professionals] lack a library with essential books to learn. It would be a great help, but unfortunately, we are completely abandoned."*

Primary health care providers are required to refer patients with health conditions that are beyond their scope of practice to general/comprehensive hospitals. However, these providers make decisions on their own, despite having limited competence. Moreover, the interviewed health professionals reported that supportive supervision, merit based performance appraisal and support from the health system are critical for their performance. A 32-year health professional reported that *"We [primary health care professionals] require support [through] supportive supervision. It would be beneficial to have a supervisor spend sufficient time with us to identify what works and what doesn't, that is not actually happening in our health center."*

The respondents also reported that regulations, policies, strategies, directives of health workforce are not translated into practice, they exist only on papers. A 37-years old health professional with a work experience of 12 years reported that *"doing based on lies has become a new normal and organizational culture in different levels of the health system. You will get only your salary whether you are competent or not. Still, a certificate of work experience regardless of your competence is the only thing required for renewal of license. The health system neither properly pays nor regulates the health professionals, especially at primary health care units where the poor and rural communities get the health care services with no other options; this is the result of poor health leadership and governance."*

## Predictors of health professionals' competence

Age, educational status, in-service training, licensure exam/COC, higher education institutions, cumulative GPA and monthly salary were statistically associated with health professionals' competence as per the global competency framework for universal health coverage.

A one-year increase in age results in a 15% decline in health professionals' competence score to provide quality primary health care. A unit increase in cumulative GPA makes health professionals to be 4.2 times more competent for providing quality primary health care. Degree holders' were almost 2 times more likely to be competent than those health professionals with diploma educational status. Health professionals who took at least one in-service

training were 2.5 times more likely to be competent than those health professionals with no in-service training after graduation. Health professionals who took the national licensure exam/COC were 2.2 times more likely to be competent than those health professionals who never took the national licensure exam/COC. Health professionals who were graduated from public higher education institutions were 4.56 times more likely to be competent than those of private colleges graduates. Moreover, health professionals with above 10,000 ETB (177.84$) monthly salary were almost 3 times more likely to be competent than those earning below 5000 ETB (88.9$) (Table 3).

## Discussion

This research aimed at assessing the state of health professionals' competence and associated factors for the provision of quality primary health care, using the world health organization's global competency framework for universal health coverage, in Amhara region, Ethiopia.

Only 116(14%) health professionals were competent as per global competency framework for universal health coverage. Similarly, most interviewed health professionals admitted their competence gaps to provide quality primary health care. Educational status, in-service training, taking licensure/COC exam, learning in public higher education institutions, high cumulative GPA and monthly salary above 10,000 ETB positively affect health professionals' competence, whereas age negatively affects health professionals' competence.

This study implies that low competence level of health professionals compounded with their shortage is a major obstacle to achieving universal health coverage through PHC in Ethiopia in general, in Amhara region in particular. This is due to the fact that on one hand, the working and living environment of PHC settings is not conducive to attracting and retaining those health professionals who are competent to provide quality PHC. They lack

**Table 3. Factors associated with health professionals' competence, Amhara region, Ethiopia, 2024.**

| | | Competent | | | | |
| | | Yes | No | | | |
| Variables | Categories | Number | Number | COR (95% CI) | AOR (95% CI) | P-value |
|---|---|---|---|---|---|---|
| Age (in years) | | | | 0.92 (0.89,0.96) | 0.85(0.8–0.90) | <0.001 |
| Cumulative GPA | | | | 7.831(4–15.33) | 4.2(2.1–8.53) | <0.001 |
| Sex | Female | 37 | 328 | 1 | 1 | |
| | Male | 79 | 386 | 1.81(1.2–2.754) | 1.22(0.73–2.03) | 0.455 |
| Educational status | Diploma | 22 | 257 | 1 | 1 | |
| | Degree | 94 | 457 | 2.40(1.47–3.92) | 1.88(1.06–3.34) | 0.032 |
| Taking IST | No | 15 | 215 | | 1 | |
| | Yes | 101 | 499 | 2.90(1.65–5.11) | 2.51(1.36–4.64) | 0.003 |
| Taking NLE | No | 22 | 268 | 1 | 1 | |
| | Yes | 94 | 446 | 2.567(1.58–4.18) | 2.23(1.29–3.84) | 0.004 |
| HEI | Public | 99 | 433 | 3.779(2.21–6.46) | 4.56(2.42–8.62) | <0.001 |
| | Private | 17 | 281 | 1 | 1 | |
| Health facility | Primary hospital | 13 | 87 | 0.91(0.49–1.69) | 0.69(0.33–1.44) | 0.327 |
| | Health center | 103 | 627 | 1 | 1 | |
| Monthly salary | <5000ETB(88.9$) | 11 | 119 | 1 | 1 | |
| | 5000–10,000ETB(88.9–177.84$) | 87 | 513 | 1.84(0.95–3.54) | 1.3(0.6–2.8) | 0.514 |
| | >10,000ETB(177.84$) | 18 | 82 | 2.38(1.07–5.291 | 3(1.1–8.49) | 0.039 |

*IST: in-service training; NLE: national licensure exam and HEI, higher education institution.*

fair payment, proper equipment, PHC-tailored quality pre-service education, decent work and security in these settings. On the other hand, nongovernmental organizations offer good payments, proper equipment, more conducive working and living environment, and appropriate mentoring and coaching, among others, which pull the competent health professionals away from PHC facilities. Therefore, this research implies that the government does not fulfill the promise of PHC on the ground, since nothing can be done there without competent health professionals.

The proportion of competent health professionals in the current study, 14%, is much lower than that of previous studies done on competence of specific health professionals' groups (nurse, midwives) [29,31,32] or specific competencies (cultural competence, digital competence)[33, 34]. The possible reason for this discrepancy might be due to the fact that the current study used WHO's global competency framework that includes new emerging contemporary competencies (e.g., putting people at the center of all practices, systems thinking, critical thinking, creativity, lifelong learning and strategic thinking) [1,7] which were not be considered in the previous studies. These competencies may not include in most of curricula of health professionals' educational programs despite they are critical for providing integrated people centered care. Moreover, the current study only includes health professionals working in PHC facilities such as primary hospitals and health centers. These professionals live and work in areas without internet access, functional libraries, learning forum, in adequate supportive supervision, coaching and mentoring. Some of the health facilities had no electricity, mobile network coverage, and are located far away from towns without infrastructure such as roads. These factors create barriers for maintaining and updating their competencies with different platforms in these geographically isolated health facilities resulting in lower level of professional competence.

The finding from the current study revealed that upward training (from diploma to degree), in-service training, taking licensure/COC exam, learning in public higher education institutions, high cumulative GPA and monthly salary above 10,000 ETB positively affect health professionals' competence which is consistent with that of previous studies [27,29,33,43]. In contrast with findings from previous studies [44, 45], in the current study, work experience had no effect on health professionals' competence. This might be due to work experience alone is not sufficient to improve competence, and that other factors such as ongoing training and education are necessary to maintain and improve competence. Evidence showed that competencies can be lost if health professionals are not lifelong learner through face to face or virtual platforms [1,7,40].

The qualitative findings of the current study also revealed some problems related to the competence of health professionals, such as: providing health care beyond their scope of practice; failing to identify which patients can be managed in PHC facilities and which ones need timely referrals to referral hospitals for specialty services. Most of the respondents acknowledged their competency gaps and expressed their willingness to update their skills if they could access training and CPD with the necessary resources. This may be seen as an opportunity to improve their competence. At least the health professionals clearly know their problems, which is half of the solution. According to the WHO, most (90%) essential health services of UHC can be addressed with PHC [46], but without competent health professionals, PHC is an empty promise on the ground. Primary health care facilities are the only, or the main, health service providers in rural, hard to reach and underserved populations with no other option. This leaves many poor people behind.

Although the purpose and process of the study were clearly explained to the study participants with the aim of getting their honest information, this study assessed health professionals' competence with self-rated items, which may understate or exaggerate their competence.

## Conclusions

The proportion of competent health professionals was very low as per global competency framework for universal health coverage and most interviewed health professionals admitted their competence gaps and expressed their desire to update their competence if they had access to the opportunities to do so. Upward training (from diploma to degree), in-service training, taking licensure exam/COC, learning in public higher education institutions, high cumulative GPA and monthly salary above 10,000 ETB positively affect health professionals' competence, whereas age negatively affects health professionals' competence.

## Recommendations

### For Amhara regional health bureau

- Implement need based CPD and linking with renewal of license as per the national CPD roadmap and the regional health workforce strategic plan (2023–2030)

- Promote and expanding of CPD centers in the region

- Promote upgrading of diploma health professionals at least to degree level education status

- Promote need based in-service training across all primary hospitals and health centers

- Deploy health professionals who pass licensure exam with high CGPA, graduated from public higher education institution at primary hospitals and health centers

- Asses the competence of health professionals at least once in a year and take evidence based action

- Should strengthen regulation of health professionals

- Promote health professionals' salary scale revision based on the inflated market

### For higher education institutions found in Amhara region

- Include the competencies that are identified in the global competency framework for UHC to educational curricula

- Promote the national licensure exam in collaboration with EMOH

- Support students to score high CGPA at the time of graduation

### For policymakers

- Policymakers should translate polices, regulations, strategic plans and directives related to health professionals' competence in to action.

### For researchers

- Develop tailored national competency framework and entrustable professional activities (EPAs)for all health professionals' educational programs

## Acknowledgments

We acknowledge the Amhara region and the study health facilities for their permission to conduct the study. We also acknowledge the study participants for providing information, the data collectors and the supervisors for collecting the data properly.

## Author contributions

**Conceptualization:** Gebeyehu Tsega, Mirkuzie Woldie, Getu Degu.

**Data curation:** Gebeyehu Tsega, Mirkuzie Woldie, Getu Degu.

**Formal analysis:** Gebeyehu Tsega, Mirkuzie Woldie.

**Funding acquisition:** Gebeyehu Tsega.

**Investigation:** Gebeyehu Tsega, Mirkuzie Woldie, Getu Degu.

**Methodology:** Gebeyehu Tsega, Gizachew Yismaw.

**Project administration:** Gebeyehu Tsega, Getu Degu.

**Resources:** Gebeyehu Tsega.

**Software:** Gebeyehu Tsega.

**Supervision:** Gebeyehu Tsega, Mirkuzie Woldie, Gizachew Yismaw, Getu Degu.

**Validation:** Gebeyehu Tsega.

**Visualization:** Gebeyehu Tsega, Mirkuzie Woldie.

**Writing – original draft:** Gebeyehu Tsega.

**Writing – review & editing:** Gebeyehu Tsega, Mirkuzie Woldie, Gizachew Yismaw, Getu Degu.

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
