## [Decision Letter · Decision Letter 0]

5 Nov 2024

PONE-D-24-20430Health professionals’ competence for the provision of quality primary health care in Amhara region, EthiopiaPLOS ONE

Dear Dr. Nebeb,

Thank you for submitting your manuscript to PLOS ONE. After careful consideration, we feel that it has merit but does not fully meet PLOS ONE’s publication criteria as it currently stands. Therefore, we invite you to submit a revised version of the manuscript that addresses the points raised during the review process. Please submit your revised manuscript by Dec 20 2024 11:59PM. If you will need more time than this to complete your revisions, please reply to this message or contact the journal office at plosone@plos.org . Please include the following items when submitting your revised manuscript:

We look forward to receiving your revised manuscript.

Kind regards,

Mulu Tiruneh

Academic Editor

PLOS ONE

Journal Requirements:

2. We note that your Data Availability Statement is currently as follows: “All relevant data are within the manuscript and in Supporting Information files.”

Please confirm at this time whether or not your submission contains all raw data required to replicate the results of your study. Authors must share the “minimal data set” for their submission. PLOS defines the minimal data set to consist of the data required to replicate all study findings reported in the article, as well as related metadata and methods (https://journals.plos.org/plosone/s/data-availability#loc-minimal-data-set-definition). For example, authors should submit the following data: - The values behind the means, standard deviations and other measures reported; - The values used to build graphs; - The points extracted from images for analysis. Authors do not need to submit their entire data set if only a portion of the data was used in the reported study. If your submission does not contain these data, please either upload them as Supporting Information files or deposit them to a stable, public repository and provide us with the relevant URLs, DOIs, or accession numbers. For a list of recommended repositories, please see https://journals.plos.org/plosone/s/recommended-repositories. If there are ethical or legal restrictions on sharing a de-identified data set, please explain them in detail (e.g., data contain potentially sensitive information, data are owned by a third-party organization, etc.) and who has imposed them (e.g., an ethics committee). Please also provide contact information for a data access committee, ethics committee, or other institutional body to which data requests may be sent. If data are owned by a third party, please indicate how others may request data access.

Reviewers' comments:

Reviewer's Responses to Questions

**Comments to the Author**

1. Is the manuscript technically sound, and do the data support the conclusions?

Reviewer #1: Yes

Reviewer #2: Yes

2. Has the statistical analysis been performed appropriately and rigorously? 

Reviewer #1: No

Reviewer #2: Yes

3. Have the authors made all data underlying the findings in their manuscript fully available?

Reviewer #1: Yes

Reviewer #2: Yes

4. Is the manuscript presented in an intelligible fashion and written in standard English?

Reviewer #1: No

Reviewer #2: Yes

5. Review Comments to the Author

Reviewer #1: Author should explain more convincingly the sample technique. Apart from this, another limitation with the competency assessment technique. Competency assessment relying upon the self reported questionnaire can only satisy the Knowledge component of competency, in that case what about attitude and skills.

Reviewer #2: The article is based on the World Health Organisation's (WHO) Global Competency Framework for Universal Health Coverage (GCUHC), and the research is closely related to international topics, which to some extent reflects the current issues facing health system challenges. The article explores the primary health care capacity of health professionals in Amhara Region, Ethiopia. A mixed-methods research design was used, including both quantitative and qualitative data, and the methods of data collection and analysis, including logistic regression modelling and thematic analysis, as well as audio-recording and cross-checking were described in detail to increase the credibility of the findings. The article's findings show that only 14 per cent of health professionals consider themselves to have all competency areas, revealing important gaps in professional competence. The article's results point to the fact that primary health care competencies are strongly associated with factors such as educational status, training, taking licensure/competency certificate exams, and monthly salary, findings that are critical for developing strategies for improvement. The article also suggests that this study addresses contemporary emerging competency issues, that there was a lack of self-perception on the part of the respondents, and that the results may not be accurate.

Recommendations:

Firstly, for the qualitative data sample of 12 technical professionals, the authors need to be mindful of whether more detailed information is provided on the qualitative data collection and analysis methods, including semi-structured questions for the interviews and specific steps for data analysis.

Secondly, the study relied on health professionals' self-assessments to measure competence, which may be subject to social desirability bias, and participants may have overestimated or underestimated their competence. It is recommended to add a third-party perspective to the assessment results.

Third, the results of the article showed that only 14% of health professionals considered themselves to have all competency domains, and the results section should discuss the specific scores and distribution of each competency domain in more detail.

Fourth, the discussion section suggests listing the consistency and discrepancy of the results with global and regional health policies.

6. PLOS authors have the option to publish the peer review history of their article (what does this mean? ). If published, this will include your full peer review and any attached files.

**Do you want your identity to be public for this peer review?** For information about this choice, including consent withdrawal, please see our Privacy Policy .

Reviewer #1: **Yes: ** Sanjeev Kumar

Reviewer #2: No

---

## [Author Response · Author response to Decision Letter 1]

11 Nov 2024

Dear Editor and Reviewers,

Thank you so much for your kind words and valuable comments, as well as the opportunity to revise the manuscript. Your feedback has greatly improved our work. We have revised the manuscript according to your suggestions and recommendations.

Reviewer #1: Author should explain more convincingly the sample technique. Apart from this, another limitation with the competency assessment technique. Competency assessment relying upon the self reported questionnaire can only satisy the Knowledge component of competency, in that case what about attitude and skills.

Response: Thank you for your kind words and insightful comment. The sampling technique is described in lines 166-173. The competence of health professionals was measured using 30 competency statements that integrated knowledge, attitude, and skill components. This method aimed to assess the health professionals' perceptions of their performance on each competency described in the global competency framework for universal health coverage (WHO, 2022). Understanding their perceptions can inform policies to address competency gaps. They rated their competence and identified gaps in each statement, indicating a need for training. In the global competency framework for universal health coverage, "competent" refers to an individual’s ability to perform designated practice activities to a defined standard, which means possessing the requisite competencies. Competencies are defined as the abilities to integrate knowledge, skills, and attitudes in performing tasks in a given context (WHO,2022). Health professionals were asked whether they performed each competency statement by integrating these three components, not just knowledge. Moreover, this limitation is described in the discussion section of the manuscript, in lines 429-431.

Reviewer #2: The article is based on the World Health Organisation's (WHO) Global Competency Framework for Universal Health Coverage (GCUHC), and the research is closely related to international topics, which to some extent reflects the current issues facing health system challenges. The article explores the primary health care capacity of health professionals in Amhara Region, Ethiopia. A mixed-methods research design was used, including both quantitative and qualitative data, and the methods of data collection and analysis, including logistic regression modelling and thematic analysis, as well as audio-recording and cross-checking were described in detail to increase the credibility of the findings. The article's findings show that only 14 per cent of health professionals consider themselves to have all competency areas, revealing important gaps in professional competence. The article's results point to the fact that primary health care competencies are strongly associated with factors such as educational status, training, taking licensure/competency certificate exams, and monthly salary, findings that are critical for developing strategies for improvement. The article also suggests that this study addresses contemporary emerging competency issues, that there was a lack of self-perception on the part of the respondents, and that the results may not be accurate.

Response: Thank you for your kind words and insightful comments. The focus is to examine health professionals' perceptions through self-rating of each competency statement/item, allowing us to hear their voices on these competencies. This self-assessment is substantiated by a qualitative exploration of their competence.

Recommendations:

Firstly, for the qualitative data sample of 12 technical professionals, the authors need to be mindful of whether more detailed information is provided on the qualitative data collection and analysis methods, including semi-structured questions for the interviews and specific steps for data analysis.

Response: Thank you for your kind words and insightful comments. We have made revisions based on your comments, as indicated in lines 225-230 and 265-272 of the revised manuscript.

Secondly, the study relied on health professionals' self-assessments to measure competence, which may be subject to social desirability bias, and participants may have overestimated or underestimated their competence. It is recommended to add a third-party perspective to the assessment results.

Thank you for your kind words and insightful comments. To address the risk of social desirability bias, we used a self-administered questionnaire in line 222, allowing respondents to rate their competence at their convenience. Additionally, the quantitative data is supported by a qualitative exploration of their competence. By recogning this, we have included this as a limitation in the discussion section of the revised manuscript, as indicated in line 429-431. Furthermore, perspectives from service users, development partners, and health managers are included in another paper as part of a Ph.D. research project. Together, these papers will provide a holistic view of health professionals' competence.

Third, the results of the article showed that only 14% of health professionals considered themselves to have all competency domains, and the results section should discuss the specific scores and distribution of each competency domain in more detail.

Response: Thank you for your kind words and insightful comments. We have revised as per the comments as indicted in lines 313-317 in the revised version along with Table 2.

Fourth, the discussion section suggests listing the consistency and discrepancy of the results with global and regional health policies.

Response: Thank you for your kind words and insightful comments. We discussed the findings in relation to Universal Health Coverage and the philosophy of primary health care, which are local, regional, and global health policies, in lines 382-383 and 424-426 of the revised version.

---

## [Decision Letter · Decision Letter 1]

26 Nov 2024

Health professionals’ competence for the provision of quality primary health care in Amhara region, Ethiopia

PONE-D-24-20430R1

Dear Dr. Nebeb,        

We’re pleased to inform you that your manuscript has been judged scientifically suitable for publication and will be formally accepted for publication once it meets all outstanding technical requirements.

Kind regards,

Mulu Tiruneh

Academic Editor

PLOS ONE

Additional Editor Comments (optional):

Reviewers' comments:

Reviewer's Responses to Questions

**Comments to the Author**

1. If the authors have adequately addressed your comments raised in a previous round of review and you feel that this manuscript is now acceptable for publication, you may indicate that here to bypass the “Comments to the Author” section, enter your conflict of interest statement in the “Confidential to Editor” section, and submit your "Accept" recommendation.

Reviewer #1: All comments have been addressed

Reviewer #2: All comments have been addressed

2. Is the manuscript technically sound, and do the data support the conclusions?

Reviewer #1: Yes

Reviewer #2: Yes

3. Has the statistical analysis been performed appropriately and rigorously? 

Reviewer #1: I Don't Know

Reviewer #2: (No Response)

4. Have the authors made all data underlying the findings in their manuscript fully available?

Reviewer #1: No

Reviewer #2: Yes

5. Is the manuscript presented in an intelligible fashion and written in standard English?

Reviewer #1: Yes

Reviewer #2: Yes

6. Review Comments to the Author

Reviewer #1: The author has adequately addressed the comments. However, i am not in position to comment on statistical analysis.

Reviewer #2: The author has already answered the previous question and there are no more questions.No other advice for this author.

7. PLOS authors have the option to publish the peer review history of their article (what does this mean? ). If published, this will include your full peer review and any attached files.

**Do you want your identity to be public for this peer review?** For information about this choice, including consent withdrawal, please see our Privacy Policy .

Reviewer #1: **Yes: ** Sanjeev Kumar

Reviewer #2: No

---

## [Editor Report · Acceptance letter]

PONE-D-24-20430R1

PLOS ONE

Dear Dr. Tsega,

I'm pleased to inform you that your manuscript has been deemed suitable for publication in PLOS ONE. Congratulations! Your manuscript is now being handed over to our production team.

Kind regards,

on behalf of

Mr. Mulu Tiruneh

Academic Editor

PLOS ONE